# Content Determination and Release Characteristics of Six Components in the Different Phases of “*Glycyrrhiza*
*glabra*-*Nux vomica*” Decoction by UPLC-MS/MS

**DOI:** 10.3390/molecules27196180

**Published:** 2022-09-21

**Authors:** Yuyan Guo, Shuang Shao, Wenjun Zhang, Chuncheng Li, Zejun Meng, Shuang Sun, Dayu Yang, Shaowa Lü

**Affiliations:** 1Key Laboratory of Chinese Materia Medica (Ministry of Education), Heilongjiang University of Chinese Medicine, Harbin 150040, China; 2Harbin University of Commerce, Harbin 150040, China; 3Daxing’anling Vocational College, Daxing’anling 165000, China

**Keywords:** decoction, phase state, *Glycyrrhiza glabra*, *Nux vomica*, content determination, in vitro release

## Abstract

The decoction turns into a complex multiphase system following exposure to high temperature and a complex chemical environment. However, the differences in the concentration of key active ingredients in different phase states and the release of drugs in sedimentary phase have yet to be elucidated. A simple ultra-performance liquid chromatography–tandem mass spectrometry (UPLC-MS/MS) method was developed and validated for the simultaneous quantitative determination of brucine, strychnine, liquiritin, isoliquiritin, isoliquiritigenin and glycyrrhizic acid concentrations and it was applied to compare the content of different phases and measure the release characteristics of the sedimentary phase in “*Glycyrrhiza glabra*-*Nux vomica*” decoction (NGD). The results show that the method’s selectivity, precision (intraday and interday ≤ 2%), matrix effect (101–108%), recovery and stability results were acceptable according to the guidelines. The method is sensitive and reliable. The content determination results show that the most toxic strychnine in the sedimentary phase accounted for 75.70% of the total components. The different components exhibited differential release in different media, and its components were released in the artificial intestinal fluid up to 81.02% in 12 h. Several components conformed to the primary kinetic model and the Ritger–Peppas model, and the most toxic compound exhibited slow release, thus conforming to the Ritger–Peppas model. This study provides a standard of reference for studies investigating reduction in toxicity of the combination of *Glycyrrhiza glabra* (*Glycyrrhiza glabra* L.) and *Nux vomica* (*Strychnos nux-vomica* L.).

## 1. Introduction

Decoction is one of the oldest and most common forms of administration of herbal medicine in China. Chinese herbs with medicinal properties, such as *Aconitum* (*Aconitum carmichaeli Debx* L.) [1,2] and *Nux vomica* (*Strychnos nux-vomica* L.) [3,4], are often combined to prepare tonics to reduce toxicity and increase effectiveness [5,6,7]. Modern science has found that the high temperature and complex chemistry of the decoction process, together with the dissolved secondary metabolites, including small molecules, biological macromolecules and inorganic elements, alter the physicochemical properties of herbal preparations. These natural substances, which act as suspension aids, flocculants and antiflocculants, turn the decoction into a multiphase integrated system of true solution, colloid solution, mixed suspension and sedimentary at the same time [8]. For example, a nanophase state of Baihu Tang was reported [9]. The study confirmed that herbs containing phenolic and carboxylic acid components, when combined with herbs containing alkaloid components, generate new compounds, which are obtained as precipitates and deposits during decoction [10]. Jinming et al. found that the sedimentary phase was generated by combinations of *Glycyrrhiza glabra* (*Glycyrrhiza glabra* L.) and *Aconitum*, which decreased its toxicity [11]. Despite the progress achieved, the composition, drug release behavior and gastrointestinal absorption of drugs in different phase states of herbal medicine decoctions show variation. Further studies analyzing the distribution and release of toxins (potent) in each phase are needed to determine the phase characteristics, but also provide a standard of reference for further exploration of the mechanism of toxicity reduction and potency enhancement of traditional Chinese medicines (TCM).

As a typical toxic TCM, *Nux vomica* (NV) has been recommended by physicians because of its remarkable pharmacological activity [12,13,14]. However, due to their inherent toxicity associated with brucine and strychnine, careless use of NV can result in poisoning and even death. Excessive use of NV can overstimulate medulla oblongata and result in neurotoxicity [15]. In order to effectively reduce the toxicity of NV, it is combined with *Glycyrrhiza glabra* (GG). GG is an ancient TCM and is an antidote against poisoning [16,17,18,19]. GG can be used to ameliorate the toxicity of NV. Many ancient books recorded the use of “raw NV plus GG immersion”, administered orally as a prescription of NV combined with GG to hydrolyze NV. The combined intake of NV and GG can reduce toxicity. However, the phase characteristics of “*Glycyrrhiza glabra*-*Nux vomica*” decoction (NGD) have yet to be analyzed comprehensively. The correlation between the content distribution in different phases and the reduction of NV toxicity using GG has yet to be explored.

It has been found that TCM prescriptions containing flavonoids and alkaloids precipitate significantly in the process of decocting [20]. For example, liquiritin and glycyrrhizic acid in Sini decoction and Mahuang-Fuzi-Gancao decoction can combine with alkaloids in Aconitum aconite to form insoluble salts. Some studies [21] have also shown that alkaloids in GG and Aconitum can form molecular complexes, which can release toxic components slowly in the body. Further investigation revealed that the formation mechanism of composite deposition was the association of tertiary amine N in alkaloids with carboxylic acid C=O in GG, which avoided the toxic reaction caused by excessive alkaloids absorbed by the organisms in a short time. Based on the above, the main alkaloids that contribute more to the formation of phase state, brucine and strychnine, and the characteristic flavonoids and carboxylic acid components of GG, liquiritin, isoliquiritin, isoliquiritigenin and glycyrrhizic acid, were selected as indexes to investigate the content distribution and release behavior of phase states.

At present, the HPLC method is still the main method for the determination of the compatible components of GG and NV. However, it is difficult to sensitively determine the content of substances in the phase states with lower concentration. In this study, centrifugation and dialysis were used to separate the phase states of NGD. The particle size and potential of different phase states, including the sedimentary phase group (NG1), suspension phase group (NG2), colloid phase group (NG3) and solution phase group (NG4) were characterized. A simple, sensitive and selective UPLC-MS/MS method was developed to determine the concentration of six components, namely brucine, strychnine, liquiritin, isoliquiritin, isoliquiritigenin and glycyrrhizic acid, in different phase states of NGD, and to analyze the differences in the distribution of the key ingredients in different phase states. Further in-depth analysis of the release behavior of NG1 revealed a relatively high proportion of macromolecules. The study showed that the distribution of compounds in different phase states varied significantly, and NG1 was the key phase containing enriched active ingredients and manifested a slow release of soluble skeleton material. The results provide a standard of reference for the study of toxicity reduction by combining NV and GG.

## 2. Results

### 2.1. Method Development and Optimization

To ensure enhanced analyte sensitivity and signal strength, the positive and negative ion modes of the measured components were selected, respectively. Brucine and strychnine easily exhibit a quasi-molecular ion peak of [M + H]^+^ with increased responsiveness under positive ion mode. In contrast, the four components in licorice readily showed [M − H]^+^ quasi-molecular ion peaks with higher responsiveness when a negative ion mode was detected. The optimized ion pair in MRM mode can be used for quantification. The precursor and product ions of brucine, strychnine, liquiritin, isoliquiritin, isoliquiritigenin and glycyrrhizic acid were *m*/*z* 395.09→243.88, *m*/*z* 335.06→156.00, *m*/*z* 417.12→255.07, *m*/*z* 417.12→255.06, *m*/*z* 255.07→119.05 and *m*/*z* 821.40→351.06, respectively. The structures and mass spectra of brucine, strychnine, liquiritin, isoliquiritin, isoliquiritigenin and glycyrrhizic acid are shown in Figure 1.

The mobile phase consisted of solvent A (containing 0.1% aqueous formic acid) and solvent B (acetonitrile containing 0.1% formic acid). The gradient elution procedure was as follows: 0–5 min, 81% A, 5–9 min, 81–50% A, 9 min–12 min, 50–10% A, 12 min–13 min, 10–90% A, 13 min–19 min, 90% A. The flow rate was 0.3 mL/min. The sample injection volume was 3 μL.

### 2.2. Analytical Method Validation

#### 2.2.1. Selectivity

Under the above chromatographic conditions, chromatographic peaks 1, 2, 3, 4, 5 and 6 were determined by comparing with the retention time and UV spectrum of the reference substance, corresponding to six indicator components of brucine, strychnine, liquiritin, isoliquiritin, isoliquiritigenin and glycyrrhizic acid, respectively. The retention time of test solution and reference solution was similar, and no interference peak was detected at the same position, demonstrating good specificity. According to the peak area of liquiritin, the separation degree ≥ 1.5 and theoretical plate number ≥ 4000 meet the requisite criteria, as shown in Figure 2.

#### 2.2.2. Precision

The precision of the instrument was evaluated and the results showed that the intraday and interday precisions of brucine, strychnine, liquiritin, isoliquiritin, isoliquiritigenin and glycyrrhizic acid exhibited were RSD ≤ 2%, indicating good precision.

#### 2.2.3. Calibration Curve and LOQs

Under optimal chromatographic conditions, the limit of quantification (LOQ) is obtained when the signal-to-noise ratio is 10. The solution of each reference substance was slowly diluted until the SNR was 3:1, and the limit of detection (LOD) of each reference substance was obtained. The LODs of brucine, strychnine, liquiritin, isoliquiritin, isoliquiritigenin and glycyrrhizic acid were 2.48, 2.21, 3.18, 2.90, 0.50 and 2.64 ng/mL, respectively. The LOQs were 8.28, 7.35, 7.72, 9.67, 1.64 and 8.80 ng/mL, respectively. Linear regression was carried out with sample concentration (X) as abscissa and peak area (Y) as ordinate. The following regression equations for each component were obtained (Table 1), indicating a good linear relationship.

#### 2.2.4. Repeatability

The reproducibility of the assay was evaluated. The RSD values of brucine, strychnine, liquiritin, isoliquiritin, isoliquiritigenin and glycyrrhizic acid were 0.23%, 0.18%, 0.19%, 0.25%, 1.24% and 0.59%, respectively, with RSD ≤ 2%, indicating good repeatability of the method.

#### 2.2.5. Stability

The contents of brucine, strychnine, glycyrrhizin, isoglycyrrhizin, isoglycyrrhizin and glycyrrhetinic acid were determined via stability tests in methanol, distilled water, artificial gastric juice and artificial intestinal juice at 0, 2, 4, 6, 8, 12 and 24 h. The RSD ≤ 2% indicated strong stability of the method.

#### 2.2.6. Recovery and Matrix Effect

The results (Table 2) showed that the recoveries of brucine, strychnine, glycyrrhizin, isoglycyrrhizin, isoglycyrrhizin and glycyrrhetinic acid were 100.08%, 99.85%, 100.38%, 100.94%, 99.77% and 99.58%, respectively, at low, medium and high concentrations. The RSDs were 1.68%, 1.93%, 1.54%, 1.72%, 1.99% and 1.89%, respectively. The average matrix effect of three concentrations of six analytes ranged from 101% to 108%, indicating that the six analytes in this method were unaffected by the matrix effect.

### 2.3. Content Determination

The results (Table 3) showed that the content of brucine and strychnine in NG1 accounted for 72.83% and 65.68%, respectively, in the combined decoction. The levels of liquiritin, isoliquiritin, isoliquiritigenin and glycyrrhetinic acid were 46.53%, 69.23%, 58.39% and 42.43%, respectively, in the combined decoction. The concentration of each component differed significantly from that of the other groups (*p* < 0.05). The content of strychnine, glycyrrhizin and glycyrrhetinic acid in NG2 was higher, accounting for 9.87%, 24.76% and 26.85% of their levels in the combined decoction, respectively. In NG3, the levels of brucine and strychnine were higher than those of other components, and the level of brucine was higher than that of strychnine. In NG4, the level of each component was lower than in other phases. A comparison of the concentration of the six components in different phase states is presented in Figure 3. The results indicated that most of the active components of NGD were concentrated in NG1.

### 2.4. In Vitro Release Determination

The cumulative release percentages of NG1 in distilled water, artificial gastric fluid and artificial intestinal fluid as the release media are shown in Table 4, Table 5 and Table 6. The release curves are plotted in Figure 4. The release of brucine and strychnine in artificial gastric fluid and artificial intestinal fluid was higher than in distilled water, and the highest release occurred in artificial intestinal fluid, with the release rates reaching 81.02% and 71.74% after 12 h of release, respectively. The release of liquiritin in distilled water was similar in artificial gastric fluid, and was about 70% after 12 h of release, but greater than that of artificial intestinal fluid. The release of isoliquiritin was lower in artificial intestinal fluid, reaching 55.09% after 12 h release. The release of isoliquiritigenin was the highest in artificial gastric fluid, reaching 60.51% after 12 h, compared with 46.01% and 55.01% in distilled water and artificial intestinal fluid, respectively, after similar duration. By contrast, glycyrrhetinic acid showed the lowest release in artificial gastric juice, while it was higher in water and artificial intestinal fluid, with a release of about 69.39% and 65.37% after 12 h, respectively.

### 2.5. Fitting of Drug Release Model In Vitro

The results of the in vitro release model in distilled water are shown in Table 7, which indicates that brucine, glycyrrhizic acid, liquiritin, isoliquiritin and isoliquiritigenin fit the primary kinetic model, and strychnine fits the Ritger–Peppas model. The results of the in vitro release model fits in artificial gastric fluid are shown in Table 8, with brucine, glycyrrhizic acid, liquiritin, isoliquiritin and isoliquiritigenin showing the best fit with the primary kinetic model, and strychnine fitting best with the Ritger–Peppas model. The results of the in vitro release model fit in artificial intestinal fluid are shown in Table 9, suggesting that brucine, glycyrrhizic acid, liquiritin, isoliquiritin and isoliquiritigenin fit best with the primary kinetic model, and strychnine fits best with the Ritger–Peppas model.

## 3. Discussion

Chinese medicine is essentially based on the compatibility of the effective components. The complexity of the biocompatibility is reflected by the diversity of chemical components and body reactions. “Phase state” refers to the relatively stable structure formed by the interaction between different or the same components in a certain environment. The chemical components in different phase states of the decoction exhibit different chemical states, including free, combined and complex types. The distribution of active components in different phase states directly affects their physical and chemical properties and biological activities. A large amount of sediment will appear in TCM after decoction is allowed to settle, which is defined as the sedimentary phase. The NG1, NG2, NG3 and NG4 phases in the decoction, especially in the toxic TCM decoction, carry different components. The determination and analysis of the distribution of the principal components in different phase states and the release behavior of the main phases can provide a standard of reference for the elucidation of the synergistic mechanism of toxicity reduction based on the compatibility of TCM.

Now, the content determination methods of NGD are mostly used in the HPLC method [22]. The report of liquid mass spectrometry is only limited to the research results of our research group [23,24,25]. On the basis of previous research results, the UPLC-MS/MS method was established for the determination of principal components in the NGD and its different phases. The components with lower content in NG4 and NG2 could be accurately determined. The contents of brucine and strychnine in NV were decreased after the combination with GG, and the decrease in strychnine was more obvious. Further analysis of the content distribution of the six main components in different phase states showed that the content distributions of brucine and strychnine in different phase states were in the following order: NG1, NG3, NG2, NG4. Strychnine, the most toxic component in NV, accounted for 75.7% of the NG1. The distribution of the main components in GG from large to small was as follows: NG1, NG2, NG3, NG4. Similarly, the levels of glycyrrhizic acid and the other four components in the NG1 ranged from 51.4% to 75.5%. Thus, NG1 is the key phase for the enrichment of active substances among the different phases of NGD obtained.

An in-depth study of the release of each substance in the sedimentary phase in different dissolution media and fitting analysis revealed that brucine, strychnine and glycyrrhizic acid were released in large amounts in the artificial intestinal fluid. Brucine, strychnine and glycyrrhizic acid were released in descending order because brucine and strychnine are carbazole alkaloids, which are released easily due to the decreased interaction between the molecules in a weakly alkaline environment. Glycyrrhizic acid is a triterpenoid saponin with carboxyl group and acidic structure, which is easily dissociated in alkaline environment. Results of the model-fitting studies suggest that the release behavior of the most toxic strychnine fitted best with the Ritger–Peppas model, showing the slow release of the soluble components. Combined with previous reports of glycyrrhizic acid adsorption of strychnine, sediments reacted to form water-insoluble compounds, suggesting that the toxic (effective) components, such as strychnine and glycyrrhizic acid, existed in the form of sedimentary copolymers due to their inherent dissolution characteristics, which resulted in a slow-release behavior in the body. This phenomenon is also consistent with the large number of flocculents generated in NG, reported in our previous study.

## 4. Materials and Methods

### 4.1. Chemicals and Reagents

Brucine and strychnine (purity 98%, Y16A7S13272 and Z02A7S18869) were purchased from Shanghai Yuanye Biotechnology Co. Ltd. (Shanghai, China). Glycyrrhizic acid, liquiritin and isoliquiritin (purity 98%, HI042238196, HL04223898 and HI042234198) were purchased from Baoji Chenguang Technology Co., Ltd. (Baoji, China)

NV (number 20201102) and GG (number 20200601) were purchased from Harbin Tongren Medicine Market and identified by Mr. Wang Zhenyue from Heilongjiang University of TCM. HPLC-grade methanol, acetonitrile and formic acid were purchased from Dikama (Dikama, USA). PBS solution was purchased from Beijing Boao Tuoda Technology Co., Ltd. (Biotopped, Beijing). Deionized water was prepared using a Milli-Q ultrapure water preparation instrument (Millipore, Burlington, MA, USA). Other reagents were of analytical grade. 

### 4.2. Preparation of NGD Phase States

Amounts of 100 g of NV and 300 g of GG were weighed and mixed with 5 volumes of water, heated and refluxed twice, initially for 60 min and then for 30 min, then mixed and refluxed twice and filtered. The filtrate was stored at −4 °C for 24 h. Phase separation was carried out via the following steps to obtain NG1, NG2, NG3 and NG4 phases (Figure 5).
(1)Preparation of NG1: After standing, the extract was filtered with a dust-free filter paper and the solids precipitated on the filter medium were identified as NG1.(2)Preparation of NG2: The supernatant obtained in step (1) was centrifuged at 12,000 r/min for 30 min at low temperature, and the precipitate obtained was designated as NG2.(3)Preparation of NG3: The supernatant obtained in step (2) was transferred carefully into a dialysis bag. The extract was separated into two portions, according to the method described for dialysis. The portion inside the dialysis bag was considered as NG3.(4)Preparation of NG4: The portion outside the dialysis bag in step (3) was designated as NG4.

All samples mentioned above were vacuum freeze-dried for backup. The yields in NG1, NG2, NG3 and NG4 groups were 41.78%, 2.57%, 48.51% and 7.10%, respectively. The obtained phases were analyzed using a Malvern laser particle sizer to determine the particle size and potential differences between the different samples (Figure 6).

### 4.3. Chromatographic and Mass Spectrometric Determination Conditions

An ACQUlTY ultra-performance liquid chromatographic system (Waters, Milford, MA, USA) was used. The column temperature was set to 40 °C. The mobile phase consisted of solvent A (containing 0.1% aqueous formic acid) and solvent B (acetonitrile containing 0.1% formic acid). The gradient elution procedure was as follows: 0–5 min, 81% A; 5–9 min, 81–50% A; 9 min–12 min, 50–10% A; 12–13 min, 10–90% A; 13–19 min, 90% A. The flow rate was 0.3 mL/min. The sample injection volume was 3 μL.

Mass spectrometry was performed using an AB SCIEX 4500 QTRAP-triple quadrupole linear ion trap tandem mass spectrometer (QTrap–MS/MS, AB Sciex, Foster City, CA, USA) equipped with an electrospray ionization source (ESI). A multistage reaction monitoring (MRM) mode with a switch between positive and negative ions was used; the source injection voltage was set to 3.5 kV. The ion source temperature was set to 320 °C. The pressure of the nebulizer was 35 psi. Depolymerization potential (DP) and collision energy (CE) were optimized for all the analytes. The optimized MS conditions are shown in Table 10.

### 4.4. Preparation of Standard Solutions, Samples and Negative Reference Solution

Brucine, strychnine, glycyrrhizic acid, liquiritin, isoliquiritin and isoliquiritigenin standards were precisely weighed and transferred to a 25 mL measuring bottle and dissolved in methanol to obtain a constant volume. The mixed control solution with brucine (0.276 mg/mL), strychnine (0.245 mg/mL), glycyrrhizic acid (0.222 mg/mL), liquiritin (0.193 mg/mL), isoliquiritin (0.290 mg/mL) and isoliquiritigenin (0.041 mg/mL) was obtained.

About 0.5 g each of NG1, NG2, NG3 and NG4 prepared as mentioned above were weighed accurately and mixed with 1 mL of concentrated ammonia water and 25 mL of chloroform in a conical bottle with a plug, and then sealed with ultrasound for 40 min to reduce the loss. After filtration, 10 mL of filtrate was removed accurately and dried in a water bath. Methanol was redissolved in a 10 mL flask, shaken well, and filtered through a microporous membrane (0.22 μm) to obtain the test solution. The negative reference solution without the test substance was prepared as described above.

### 4.5. Method Validation

The method was comprehensively validated for selectivity, calibration curve linearity, LLOQ, precision, matrix effect, recovery and stability according to the US FDA and Chinese Pharmacopoeia (2020) Guidelines for the Validation of Methods.

#### 4.5.1. Selectivity

The selectivity of the method was assessed by comparing the chromatograms of the negative samples without the measured components with those of all analyte samples. Six analytes showed no peaks at the retention time, which indicated lack of interference with the test samples.

#### 4.5.2. Precision

The intra- and interday precision were determined by testing the same standard solution in six replicates for three days in a row. Precision was shown by the coefficient of variation (RSD, %). The RSD values should not exceed 2%.

#### 4.5.3. Calibration Curve and LOQs

Under the optimal chromatographic conditions, a detection signal-to-noise ratio of 10 was used as the limit of quantification of the analytes. The signal-to-noise ratio of 3:1 was used to determine the detection limit of each component. Linearity was investigated by plotting the peak areas of the analytes against the measured concentrations of the standards. The calibration equation was calculated using weighted least-squares regression (weighting factor of l/χ^2^) and the correlation coefficient r was greater than 0.999.

#### 4.5.4. Repeatability

To evaluate the precision of the analytical method, six test samples were analyzed and the concentration of the components in each sample was calculated and expressed as RSD ≤ 2.0%, indicating good reproducibility.

#### 4.5.5. Stability

To evaluate the stability of the analytes during sample preparation and storage, we monitored the concentrations of six analytes under different storage conditions and over 24 h. Expressed as the mean relative deviation (RSD, %), good stability was indicated when the RSD ≤ 2.0%.

#### 4.5.6. Recovery and Matrix Effect

In order to evaluate the accuracy of the method, three test samples (high, medium and low concentrations) were prepared and the spiked recoveries were determined to ensure an accuracy greater than 95% and the RSD ≤ 2.0%. The matrix effect was determined by comparing the peak area ratio of the extracted NGD with the standard solution of the same concentration. Taking into account instrument fluctuations, matrix effects between 90% and 110% indicate that the analyte is hardly affected.

### 4.6. Content Determination

Four samples from 3 batches of NG1, NG2, NG3 and NG4, respectively, were selected, according to the optimal chromatographic conditions of sample analysis. The chromatographic peak area and the concentrations of six components in different phase states were calculated based on the composition of each phase in the raw herb.

### 4.7. In Vitro Release and Model Fitting

According to the Pharmacopoeia of the People’s Republic of China 2020 edition, 500 mL of degassed water, artificial gastric juice and artificial intestinal juice were injected into each operating container, and the temperature was adjusted to 37 ± 0.5 °C. The sedimentary phase was weighed to about 2.0 g in 3 portions, weighed accurately and transferred to containers containing water, artificial gastric juice and artificial intestinal juice. The centrifugation speed was adjusted to 100 r/min, and a 2 mL sample solution was drawn from the containers at 0.5 h, 1 h, 2 h, 4 h, 6 h, 8 h, 10 h and 12 h. Meanwhile, the same medium was added to the sample solution containing 0.22 μL of filtering agent, and the filtrate was obtained for later use. According to the selected method, the total content of brucine, strychnine, glycyrrhizic acid, liquiritin, isoliquiritin and isoliquiritigenin in the sedimentary phase was 100%. The cumulative release percentage was calculated, the release curve was drawn and the release kinetics was determined.

The cumulative drug release of the original indicator components was analyzed according to zero-order and first-order kinetics, and Higuchi and Ritger–Peppas models. The correlation equation and correlation coefficient R^2^ were obtained, and the goodness of fit was judged by R^2^.

### 4.8. Statistical Analysis

The experimental data were analyzed using SPSS 26.0 software. One-way ANOVA was used for comparison of multiple groups, and LSD test was used for comparison of two groups.

## 5. Conclusions

A rapid, simple and accurate UPLC/MS method was established for the determination of six main components in different phase states of NGD, and the release characteristics of NG1 were studied. The specificity, precision, linearity, repeatability, recovery and stability of the method was validated via various tests. Based on the perspective of microscopic ingredients, the concentrations of different phase components in the decoction were determined and the release characteristics of NG1 were analyzed. It was found that the toxic (effective) components were concentrated in NG1 after NV was combined with GG, and large concentrations of the toxic (effective) components in NG1 were released in the artificial intestinal fluid, demonstrating a slow-release behavior. The results provide a scientific basis for toxicity reduction and survival using the combination of NV and GG.

## Figures and Tables

**Figure 1 molecules-27-06180-f001:**
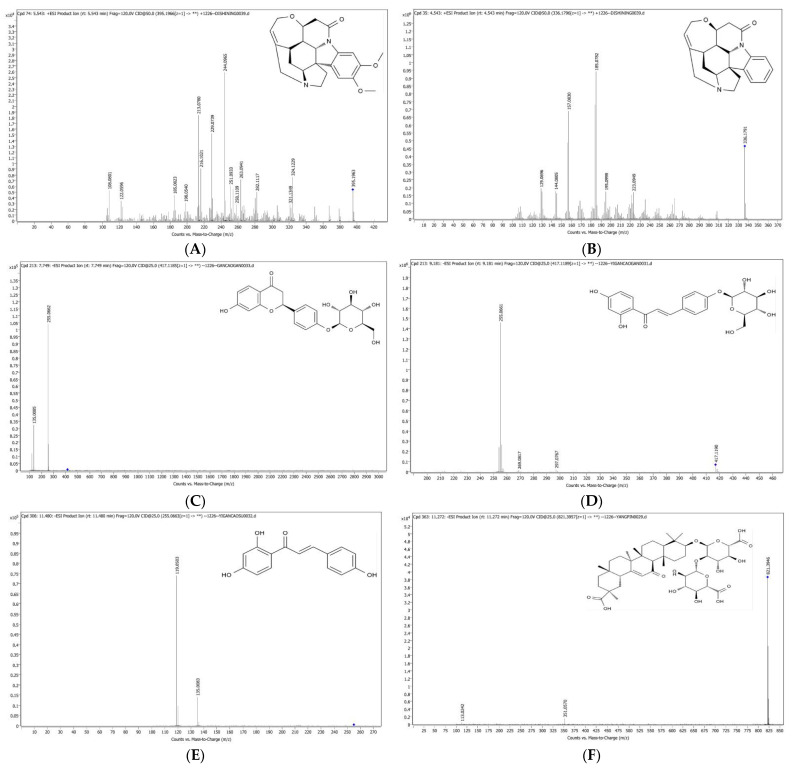
Chemical structures and mass spectra of (**A**) brucine, (**B**) strychnine, (**C**) liquiritin, (**D**) isoliquiritin, (**E**) isoliquiritigenin and (**F**) glycyrrhizic acid.

**Figure 2 molecules-27-06180-f002:**
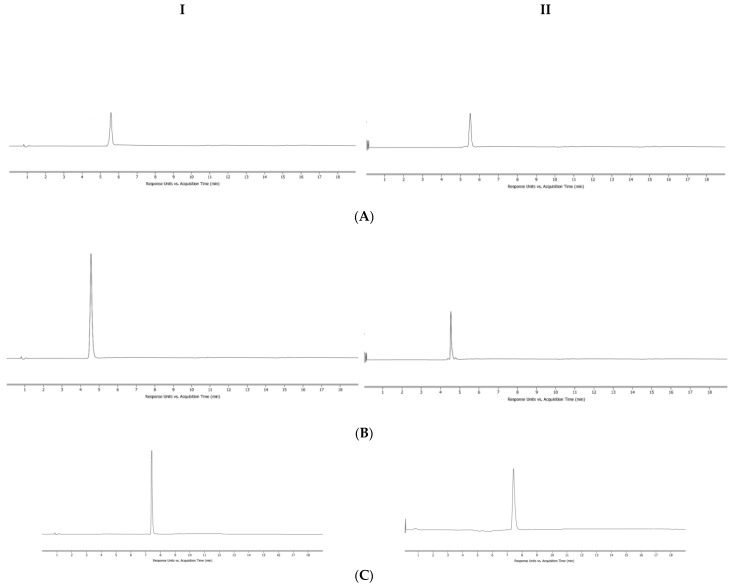
UPLC chromatogram of reference solution (**I**) and sample solution (**II**); (**A**) brucine; (**B**) strychnine; (**C**) liquiritin; (**D**) isoliquiritin; (**E**) isoliquiritigenin; (**F**) glycyrrhizic acid.

**Figure 3 molecules-27-06180-f003:**
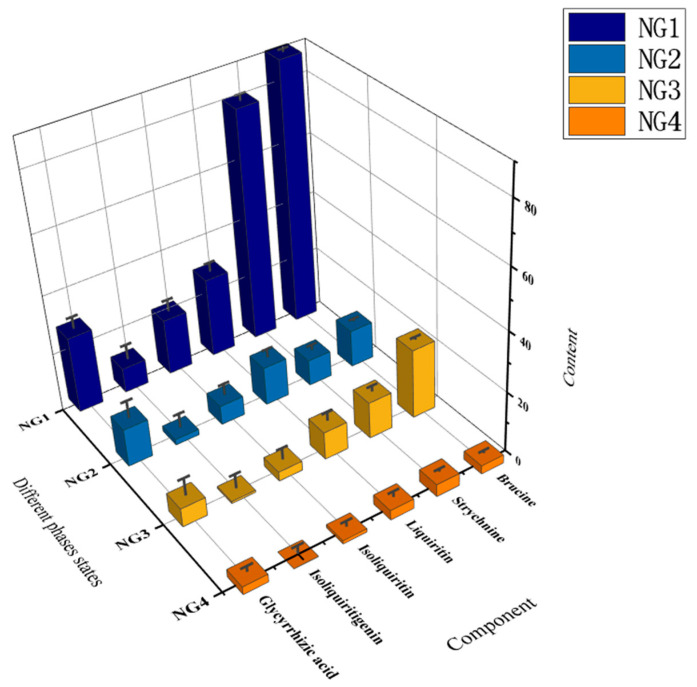
A comparison of the concentration of the six components in different phase states.

**Figure 4 molecules-27-06180-f004:**
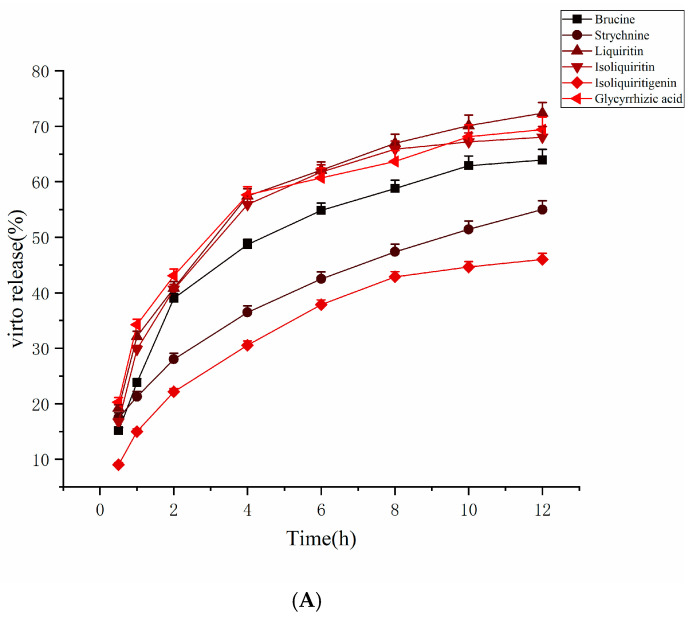
The release curve of the dissolution media; (**A**) distilled water; (**B**) artificial intestinal fluid; (**C**) artificial gastric juice.

**Figure 5 molecules-27-06180-f005:**
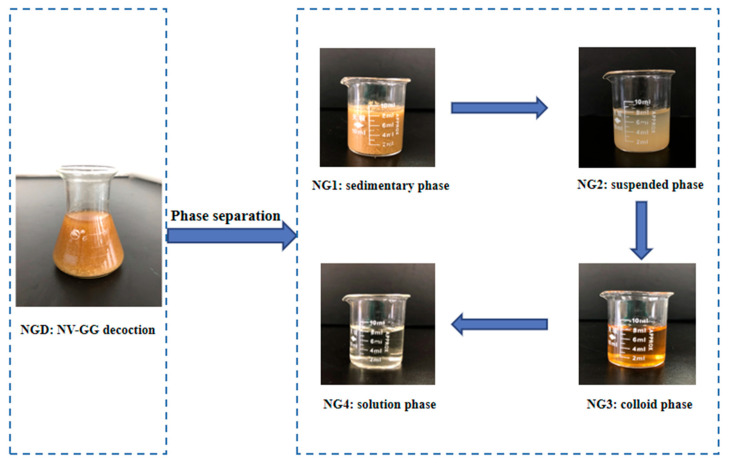
Separation of NGD phase states.

**Figure 6 molecules-27-06180-f006:**
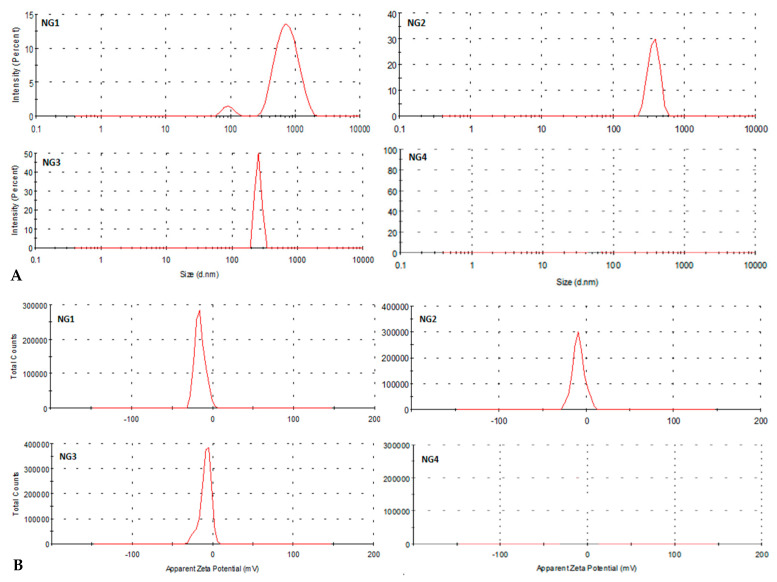
Characterization of different phase states; (**A**) particle size and (**B**) Zeta potential.

**Table 1 molecules-27-06180-t001:** Standard curves of 6 components.

Component	Calibration Curve	R^2^	Concentration Ranges (ng·mL^−1^)
Brucine	y = 0.2385x + 0.0332	0.9998	13.80~276.00
Strychnine	y = 0.2329x + 0.5199	0.9997	12.25~245.00
Liquiritin	y = 0.0898x − 0.0165	0.9997	9.65~193.00
Isoliquiritin	y = 0.1021x − 0.0718	0.9998	14.50~290.00
Isoliquiritigenin	y = 0.1699x + 0.0164	0.9996	2.05~41.00
Glycyrrhizic acid	y = 0.0974x − 0.0127	0.9998	11.10~222.00

**Table 2 molecules-27-06180-t002:** Recovery and matrix effect of 6 components.

Component	Concentration (mg/mL)	Recovery (%)	Matrix Effect (%)	RSD
		Mean ± SD (%)	Mean ± SD (%)	
Brucine	0.0182	99.07 ± 1.67	103 ± 5.29	1.68
	0.0206	99.99 ± 1.72	107 ± 3.51	
	0.0225	101.18 ± 1.49	108 ± 2.52	
Strychnine	0.0160	100.07 ± 2.32	102 ± 2.52	1.93
	0.0182	99.77 ± 2.64	101 ± 1.73	
	0.0203	99.71 ± 1.98	103 ± 4.00	
Liquiritin	0.0107	100.76 ± 1.65	106 ± 3.06	1.54
	0.0121	99.47 ± 1.15	104 ± 4.16	
	0.0121	100.90 ± 1.92	101 ± 1.73	
Isoliquiritin	0.0154	100.22 ± 2.52	104 ± 5.20	1.72
	0.0163	101.34 ± 0.66	102 ± 3.06	
	0.0182	101.60 ± 1.95	102 ± 2.31	
Isoliquiritigenin	0.0025	99.77 ± 2.61	105 ± 2.52	1.99
	0.0026	100.34 ± 2.09	102 ± 4.73	
	0.0032	99.20 ± 2.28	107 ± 5.77	
Glycyrrhizic acid	0.0158	98.65 ± 2.05	102 ± 5.51	1.89
	0.0185	100.32 ± 2.20	104 ± 3.51	
	0.0201	99.78 ± 1.73	105 ± 4.97	

**Table 3 molecules-27-06180-t003:** Determination of the concentration of each component in different phase states (mean ± SD, %).

Component	NG1 (mg/g)	NG2 (mg/g)	NG3 (mg/g)	NG4 (mg/g)
Brucine	0.882 ± 0.073 *	0.125 ± 0.013 *	0.232 ± 0.018 *	0.028 ± 0.002 *
Strychnine	0.779 ± 0.062 *	0.087 ± 0.005 *	0.122 ± 0.009 *	0.041 ± 0.002 *
Liquiritin	0.268 ± 0.031 *	0.129 ± 0.014 *	0.092 ± 0.005 *	0.032 ± 0.001 *
Isoliquiritin	0.189 ± 0.027 *	0.064 ± 0.007 *	0.031 ± 0.002 *	0.011 ± 0.001 *
Isoliquiritigenin	0.083 ± 0.002 *	0.019 ± 0.003 *	0.007 ± 0.001 *	0 *
Glycyrrhizic acid	0.258 ± 0.027 *	0.127 ± 0.011 *	0.062 ± 0.007 *	0.026 ± 0.001 *

* Compared with NG1 group, * *p* < 0.05.

**Table 4 molecules-27-06180-t004:** Percentage of cumulative release in distilled water (mean ± SD, %).

Component	0.5 h	1 h	2 h	4 h	6 h	8 h	10 h	12 h
Brucine	15.14 ± 0.58	23.82 ± 0.70	39.01 ± 0.83	48.69 ± 0.98	54.86 ± 1.29	58.8 ± 1.48	62.91 ± 1.74	63.93 ± 1.91
Strychnine	17.46 ± 0.92	21.28 ± 0.95	28.02 ± 1.08	36.48 ± 1.16	42.49 ± 1.29	47.38 ± 1.38	51.43 ± 1.52	55.01 ± 1.59
Liquiritin	19.22 ± 0.61	32.1 5± 0.90	40.82 ± 1.19	57.43 ± 1.32	62.08 ± 1.49	66.92 ± 1.68	70.14 ± 1.91	72.35 ± 1.92
Isoliquiritin	16.73 ± 0.60	29.80 ± 0.72	40.62 ± 0.88	55.89 ± 1.04	61.78 ± 1.29	65.88 ± 1.47	67.21 ± 1.59	68.02 ± 1.92
Isoliquiritigenin	9.01 ± 0.42	14.98 ± 0.53	22.16 ± 0.60	30.56 ± 0.78	37.87 ± 0.85	42.88 ± 0.91	44.65 ± 0.98	46.01 ± 1.11
Glycyrrhizic acid	20.28 ± 0.84	34.26 ± 0.97	43.10 ± 1.19	57.64 ± 1.47	60.70 ± 1.81	63.65 ± 1.98	68.11 ± 2.17	69.39 ± 2.28

**Table 5 molecules-27-06180-t005:** Percentage of cumulative release in artificial gastric juice (mean ± SD, %).

Component	0.5 h	1 h	2 h	4 h	6 h	8 h	10 h	12 h
Brucine	28.98 ± 1.37	47.47 ± 1.49	60.91 ± 1.67	66.78 ± 1.82	71.57 ± 2.09	74.62 ± 2.35	76.57 ± 2.27	78.61 ± 2.87
Strychnine	17.17 ± 0.89	22.79 ± 0.97	29.98 ± 1.05	40.25 ± 1.14	48.01 ± 1.24	54.03 ± 1.32	59.67 ± 1.47	65.03 ± 1.59
Liquiritin	18.78 ± 0.89	25.25 ± 0.97	40.85 ± 1.05	52.42 ± 1.09	59.45 ± 1.15	64.08 ± 1.21	66.34 ± 1.71	68.05 ± 2.43
Isoliquiritin	17.13 ± 0.89	26.09 ± 1.03	34.26 ± 1.08	48.64 ± 1.21	54.48 ± 1.35	58.44 ± 1.30	60.75 ± 1.57	61.55 ± 1.84
Isoliquiritigenin	8.08 ± 0.46	13.98 ± 0.56	20.47 ± 0.62	38.56 ± 0.90	44.39 ± 1.07	51.24 ± 1.35	56.44 ± 1.49	60.51 ± 1.73
Glycyrrhizic acid	6.22 ± 0.38	13.07 ± 0.42	21.64 ± 0.44	26.57 ± 0.49	30.35 ± 0.61	33.94 ± 0.80	35.97 ± 0.89	37.84 ± 0.92

**Table 6 molecules-27-06180-t006:** Percentage of cumulative release in artificial intestinal juice (mean ± SD, %).

Component	0.5 h	1 h	2 h	4 h	6 h	8 h	10 h	12 h
Brucine	30.35 ± 1.54	49.69 ± 1.59	62.23 ± 1.90	71.26 ± 2.03	71.26 ± 2.45	76.37 ± 2.84	79.63 ± 3.02	81.02 ± 3.31
Strychnine	16.58 ± 1.12	22.49 ± 1.26	30.54 ± 1.22	42.64 ± 1.37	52.01 ± 1.58	59.32 ± 1.63	65.35 ± 1.72	71.74 ± 1.85
Liquiritin	16.91 ± 0.70	25.24 ± 0.69	39.98 ± 0.73	50.22 ± 0.77	56.45 ± 1.05	63.69 ± 1.31	64.75 ± 1.60	65.94 ± 2.03
Isoliquiritin	16.88 ± 0.72	25.32 ± 0.83	31.53 ± 0.85	45.02 ± 0.94	50.82 ± 0.97	53.30 ± 0.98	54.36 ± 1.12	55.09 ± 1.37
Isoliquiritigenin	7.20 ± 0.41	14.81 ± 0.38	25.65 ± 0.43	35.60 ± 0.59	41.42 ± 0.87	46.82 ± 1.19	52.92 ± 1.24	55.01 ± 1.39
Glycyrrhizic acid	20.33 ± 0.95	28.28 ± 0.93	51.83 ± 1.10	56.79 ± 1.37	58.38 ± 1.98	60.71 ± 2.43	63.88 ± 2.71	65.37 ± 2.79

**Table 7 molecules-27-06180-t007:** Results of the cumulative release fit in distilled water.

Component	Fitting Model	Fitting Equation	R^2^
Brucine	Zero-order dynamics	Q = 24.623 + 3.912T	0.8113
	First-order dynamics	Q = 61.507 × (1 − e^−0.464T^)	0.9796
	Higuchi model	Q = 17.447 × T^0.5^ + 8.764	0.9314
	Ritger–Peppas model	Q = 26.747 × T^0.374^	0.9535
Strychnine	Zero-order dynamics	Q = 20.058 + 3.197T	0.9469
	First-order dynamics	Q = 50.543 × (1 − e^−0.420T^)	0.8787
	Higuchi model	Q = 13.762 × T^0.5^ + 8.155	0.9978
	Ritger–Peppas model	Q = 21.772 × T^0.373^	0.9994
Liquiritin	Zero-order dynamics	Q = 29.795 + 4.201T	0.8197
	First-order dynamics	Q = 68.876 × (1 − e^−0.506T^)	0.9648
	Higuchi model	Q = 18.703 × T^0.5^ + 12.836	0.9368
	Ritger–Peppas model	Q = 31.899 × T^0.349^	0.9621
Isoliquiritin	Zero-order dynamics	Q = 28.699 + 4.054T	0.7663
	First-order dynamics	Q = 66.688 × (1 − e^−0.506T^)	0.9878
	Higuchi model	Q = 18.275 × T^0.5^ + 11.849	0.9037
	Ritger–Peppas model	Q = 30.687 × T^0.352^	0.9353
Isoliquiritigenin	Zero-order dynamics	Q = 13.874 + 3.152T	0.8854
	First-order dynamics	Q = 46.164 × (1 − e^−0.316T^)	0.9827
	Higuchi model	Q = 13.816×T^0.5^ + 1.613	0.9740
	Ritger–Peppas model	Q = 15.876 × T^0.453^	0.9785
Glycyrrhizic acid	Zero-order dynamics	Q = 31.856 + 3.731T	0.7811
	First-order dynamics	Q = 65.567 × (1 − e^−0.617T^)	0.9595
	Higuchi model	Q = 16.731 × T^0.5^ + 16.535	0.9094
	Ritger–Peppas model	Q = 33.500 × T^0.313^	0.9470

**Table 8 molecules-27-06180-t008:** Results of cumulative release fitting in artificial gastric juice.

Component	Fitting Model	Fitting Equation	R^2^
Brucine	Zero-order dynamics	Q = 44.704 + 3.399T	0.6786
	First-order dynamics	Q = 74.048 × (1 − e^−0.947T^)	0.9582
	Higuchi model	Q = 15.504 × T^0.5^ + 30.194	0.8257
	Ritger–Peppas model	Q = 45.634 × T^0.236^	0.8968
Strychnine	Zero-order dynamics	Q = 20.250 + 4.021T	0.9596
	First-order dynamics	Q = 60.933 × (1 − e^−0.328T^)	0.9082
	Higuchi model	Q = 17.225 × T^0.5^ + 5.458	0.9996
	Ritger–Peppas model	Q = 22.520 × T^0.424^	0.9997
Liquiritin	Zero-order dynamics	Q = 27.168 + 4.089T	0.8164
	First-order dynamics	Q = 65.720 × (1 − e^−0.473T^)	0.9770
	Higuchi model	Q = 18.224 × T^0.5^ + 10.618	0.9355
	Ritger–Peppas model	Q = 29.277 × T^0.364^	0.9585
Isoliquiritin	Zero-order dynamics	Q = 25.167 + 3.678T	0.8219
	First-order dynamics	Q = 59.855 × (1 − e^−0.477T^)	0.9703
	Higuchi model	Q = 16.377 × T^0.5^ + 10.314	0.9396
	Ritger–Peppas model	Q = 27.021 × T^0.358^	0.9627
Isoliquiritigenin	Zero-order dynamics	Q = 12.075 + 4.530T	0.9227
	First-order dynamics	Q = 64.468 × (1 − e^−0.209T^)	0.9924
	Higuchi model	Q = 19.615 × T^0.5^ − 5.037	0.9862
	Ritger–Peppas model	Q = 15.532 × T^0.564^	0.9808
Glycyrrhizic acid	Zero-order dynamics	Q = 12.385 + 2.449T	0.8354
	First-order dynamics	Q = 36.134 × (1 − e^−0.389T^)	0.9736
	Higuchi model	Q = 10.848 × T^0.5^ + 2.614	0.9434
	Ritger–Peppas model	Q = 13.920 × T^0.419^	0.9554

**Table 9 molecules-27-06180-t009:** Results of cumulative release fitting in artificial intestinal fluid.

Component	Fitting Model	Fitting Equation	R^2^
Brucine	Zero-order dynamics	Q = 46.467 + 3.450T	0.6762
	First-order dynamics	Q = 76.146 × (1 − e^−0.972T^)	0.9535
	Higuchi model	Q = 15.716 × T^0.5^ + 31.780	0.8206
	Ritger–Peppas model	Q = 47.411 × T^0.232^	0.8921
Strychnine	Zero-order dynamics	Q = 19.642 + 4.679T	0.9622
	First-order dynamics	Q = 69.646 × (1 − e^0.272T^)	0.9385
	Higuchi model	Q = 20.018 × T^0.5^ + 2.480	0.9997
	Ritger–Peppas model	Q = 22.404 × T^0.467^	0.9998
Liquiritin	Zero-order dynamics	Q = 26.016 + 4.024T	0.8164
	First-order dynamics	Q = 64.021 × (1 − e^−^^0.465T^)	0.9752
	Higuchi model	Q = 17.931 × T^0.5^ + 9.736	0.9351
	Ritger–Peppas model	Q = 28.142 × T^0.369^	0.9573
Isoliquiritin	Zero-order dynamics	Q = 24.393 + 3.153T	0.7932
	First-order dynamics	Q = 53.770 × (1 − e^−0.528T^)	0.9618
	Higuchi model	Q = 14.123 × T^0.5^ + 11.483	0.9201
	Ritger–Peppas model	Q = 25.834 × T^0.333^	0.9503
Isoliquiritigenin	Zero-order dynamics	Q = 13.591 + 3.924T	0.8987
	First-order dynamics	Q = 55.365 × (1 − e^−0.268T^)	0.9845
	Higuchi model	Q = 17.132 × T^0.5^ − 1.533	0.9796
	Ritger–Peppas model	Q = 16.419 × T^0.503^	0.9780
Glycyrrhizic acid	Zero-order dynamics	Q = 32.820 + 3.288T	0.6403
	First-order dynamics	Q = 62.331 × (1 − e^−0.720T^)	0.9641
	Higuchi model	Q = 15.080 × T^0.5^ + 18.604	0.7917
	Ritger–Peppas model	Q = 33.952 × T^0.287^	0.8504

**Table 10 molecules-27-06180-t010:** Chromatographic and mass spectrometric parameters of MRM mode.

Analytes	Rt (min)	Precursor Ion (*m*/*z*)	Polarity	Production (*m*/*z*)	DP (V)	CE (eV)
Brucine	5.543	395.20	ESI+	244.10	100.00	45.00
Strychnine	4.543	336.18	ESI+	185.08	120.00	50.00
Liquiritin	7.749	417.12	ESI−	255.07	92.00	25.00
Isoliquiritin	9.181	417.12	ESI−	255.06	92.00	25.00
Isoliquiritigenin	11.480	255.07	ESI−	119.05	70.00	20.00
Glycyrrhizic acid	11.272	821.40	ESI−	351.06	158.00	40.00

## Data Availability

Data are contained within the article.

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
