# Peer review of "Content Determination and Release Characteristics of Six Components in the Different Phases of “Glycyrrhizaglabra-Nux vomica” Decoction by UPLC-MS/MS"

_molecules, 2022, doi:10.3390/molecules27196180_

Round 1

Reviewer 1 Report

This review  report  the UPLC-MS/MS Analysis of Six Components in the Liquid Phase of "Glycyrrhiza glabra-Nux vomica" Decoction and Their  Release Characteristics in Sedimentary Phase. The manuscript is well prepared and the subject is interesting. In my opinion, the manuscript is in a position to be accepted for publication after major revision. Here are some comments on the manuscript:

Main Observation

-       For the first citation, the binomial name (Glycyrrhiza glabra and Nux vomica) should be followed by the authority.

-       The authors must cite their novel paper concerning the same species :

Guo Yuyan, Zejun Meng, Yuanyuan Gu, Weinan Li, Shuang Sun, Qiuhong Wang, and Haixue Kuang. "HPLC-MS/MS analysis of primary alkaloids in rat plasma after oral administration of “Nux vomica-Glycyrrhiza glabra decoction”: A pharmacokinetic study." Journal of Ethnopharmacology (2022): 115588.

-       The authors should add some references to the paper. The paper is poor concerning the references (13 references only???????). 

General remarks:

-       Check the whole paper to remove spelling mistakes and insert blanks where necessary.

-       All the references should be cited in the text according to “instructions for authors” of the journal. In the same way, the references list should be revised and adapted to “instruction for authors” of the journal.

-       The titles and subtitles should be revised and adapted to “instruction for authors” of the journal.

Author Response

Dear Reviewer,

Reviewer 2 Report

Authors have developed UPLC-MS/MS method for the determination of six components of Glycyrrhiza glabra-Nux vomica decoction and their release characteristics in sedimentary phase. The proposed method was validated for several validation parameters. The manuscript is suitable for publication after some revisions suggested below:  

Abstract: The quantitative information about validation parameters is missing in the abstract. Kindly include it.

Introduction: Please describe the reported analytical methods for the determination of similar compounds and include the advantages of present method over reported analytical methods.

Figures 1, 2 and 6: The text is not clear. Please increase the size of the text.

Abbreviations: kindly define each abbreviation in its first appearance.

Discussion: Authors are advised to compare the present results with literature.

The description used to determine the "precision and accuracy" of the method needs to be improved.    

The reference is required for the standard value of percent recovery at 95% and RSD of less than 3%.

Please include system suitability parameters for the target analytical method.

The authors should include the voucher number for NV and GC.

References: The cited references are fewer in number. Authors are advised to include more references related with present work.

Author Response

Dear Reviewer,

Reviewer 3 Report

The abstract need to improve according to the comments. It is not clear the idea of the manuscript. Please improve.

In the introduction, the authors talk about how decoction is one of the oldest and most common forms of administration and their benefits and explain that in this herbal product exists a precipitated (sedimentary phase). Then, explain the health benefits of Glycyrrhiza glabra-Nux vomica and the possible metabolites related to this pharmacological activity.

Why and How selected the six metabolites which be evaluated in this study? Please, clarify this information in the introduction. Add one o two paragraphs to understand better. Remember to focus on the six metabolites.

Figure 1 has a terrible resolution, and the chemical structures are not drawn according to the ACS rules. Please improve all the components in the figure.

Why don't you evaluate the matrix effect in Analytical Method Validation? Please provide enough evidence that the matrix doesn't show a significant impact during the analysis. If your answer is that because you use MRM, please review the literature and improve this section.

In the selectivity section, please explain why you use HPLC-UV if you have a piece of better equipment coupled with MS? And provide the wavelength of each chromatogram. Add this information in Figure 2. Again, Figure 2 has a terrible resolution. Please improve this issue with professional software.

When the authors say they have good precision, please provide the reference and international guides.

In table 2, What are the unists (mg/g, mg/mL)?

Please attend to all the comments to accept the manuscript.

Author Response

Dear Reviewer,

Round 2

Reviewer 2 Report

After reconsideration of the revised manuscript, I found that the authors have responded to all comments and questions.  Thus, I am pleased to accept this manuscript for publication. 

Reviewer 3 Report

Dear authors, thank you so much for attending to my comments. 

Now the abstract presents a complete idea about the work and shows significant results.  

The introduction is complete, and thanks for adding the paragraphs to understand why you select these metabolites. 

The chemical structures in Figure 1 are correct and according to ACS rules. 

The manuscript presents a full validation protocol according to international guidelines. 

The conclusion is according to the results. 

The authors develop and present excellent and exciting work. Furthermore, they attend to and improve the manuscript according to all comments. For that reason, I consider this manuscript to publish in Molecules journal.